# General Invertible Transformations for Flow-based Generative Modeling

**Jakub M. Tomczak** [1]

## Abstract

In this paper, we present a new class of invertible transformations with an application to flow-based generative models. We indicate that many well-known invertible transformations in reversible logic and reversible neural networks could be derived from our proposition. Next, we propose two new coupling layers that are important building blocks of flow-based generative models. In the experiments on digit data, we present how these new coupling layers could be used in Integer Discrete Flows (IDF), and that they achieve better results than standard coupling layers used in IDF and RealNVP.

## 1. Introduction

**Notation** Let us consider a $D$-dimensional space $\mathbf{x} \in \mathcal{X}$, e.g., $\mathcal{X} = \{0,1\}^D$, $\mathcal{X} = \mathbb{Z}^D$ or $\mathcal{X} = \mathbb{R}^D$. We define a binary invertible operator $\circ : \mathcal{X} \times \mathcal{X} \to \mathcal{X}$. The inverse operation to $\circ$ is denoted by $\bullet$. For instance, for the addition: $\circ \equiv +$ and $\bullet \equiv -$, and for the XOR operator: $\circ \equiv \oplus$ and $\bullet \equiv \oplus$. Further, we use the following notation: $\mathcal{X}_{i:j}$ is a subset of $\mathcal{X}$ corresponding to variables from the $i$-th dimension to the $j$-th dimension, $\mathbf{x}_{i:j}$, we assume that $\mathcal{X}_{1:0} = \emptyset$ and $\mathcal{X}_{n+1:n} = \emptyset$

**Invertible transformations** In reversible computing (Toffoli, 1980; Fredkin & Toffoli, 1982), invertible logic gates allow inverting logic operations in order to potentially decrease energy consumption of computation (Bennett, 2003). Typical invertible logic gates are:
– *the Feynman gate*: For $\mathbf{x} \in \{0,1\}^2$, the gate is defined as follows (Feynman, 1986):

$$y_1 = x_1$$
$$y_2 = x_1 \oplus x_2. \tag{1}$$

[*]Equal contribution [1]Department of Computer Science, Vrije Universiteit Amsterdam, Amsterdam, the Netherlands. Correspondence to: Jakub M. Tomczal <j.m.tomczak@vu.nl>.

Third workshop on *Invertible Neural Networks, Normalizing Flows, and Explicit Likelihood Models* (ICML 2021). Copyright 2021 by the author(s).

– *the Toffoli gate*: For $\mathbf{x} \in \{0,1\}^3$, the gate is defined as follows (Toffoli, 1980):

$$y_1 = x_1$$
$$y_2 = x_2 \tag{2}$$
$$y_3 = x_3 \oplus (x_1 \wedge x_2).$$

– *the Fredkin gate*: For $\mathbf{x} \in \{0,1\}^3$, the gate is defined as follows (Fredkin & Toffoli, 1982):

$$y_1 = x_1$$
$$y_2 = x_2 \oplus (x_1 \wedge (x_2 \oplus x_3)) \tag{3}$$
$$y_3 = x_3 \oplus (x_1 \wedge (x_2 \oplus x_3)).$$

Invertible transformations play also a crucial role in reversible neural networks (Gomez et al., 2017; Chang et al., 2018; MacKay et al., 2018). For instance, an invertible transformation called a *coupling layer* is an important building block in flow-based models (Dinh et al., 2016). It is defined as follows:

$$\mathbf{y}_1 = \mathbf{x}_1$$
$$\mathbf{y}_2 = \exp\{\mathrm{NN}_s(\mathbf{x}_1)\} \odot x_2 + \mathrm{NN}_t(\mathbf{x}_1), \tag{4}$$

where $\odot$ is an element-wise multiplication, $\mathrm{NN}_s(\cdot)$ and $\mathrm{NN}_t(\cdot)$ denote arbitrary neural networks, and the input is divided into two parts, $\mathbf{x} = [\mathbf{x}_1, \mathbf{x}_2]$, e.g., along the channel dimension. If $\mathrm{NN}_s(\cdot) \equiv 1$, and we stack two coupling layers with reversing the order of variables in between, then we obtain the reversible residual block (Gomez et al., 2017):

$$\mathbf{y}_1 = \mathbf{x}_1 + \mathrm{NN}_{t,1}(\mathbf{x}_2)$$
$$\mathbf{y}_2 = \mathbf{x}_2 + \mathrm{NN}_{t,2}(\mathbf{y}_1). \tag{5}$$

Recently, (Hoogeboom et al., 2019) proposed a modification of the coupling layer for integer-valued variables:

$$\mathbf{y}_1 = \mathbf{x}_1$$
$$\mathbf{y}_2 = \mathbf{x}_2 + \lfloor \mathrm{NN}_t(x_1) \rceil, \tag{6}$$

where $\lfloor \cdot \rceil$ denotes the rounding operation. In order to allow applying bakcpropagation to the rounding operation, the straight through gradient estimator is used (Hoogeboom et al., 2019).

**Flow-based generative models** There are three major groups of generative models: autoregressive models (Van den Oord et al., 2016), latent variable models(Goodfellow et al., 2014; Kingma & Welling, 2013; Rezende et al., 2014), and flow-based models (Papamakarios et al., 2019). The last approach takes advantage of the change of variables formula:

$$p(\mathbf{x}) = \pi(\mathbf{z} = f^{-1}(\mathbf{x}))|\mathbf{J}_f(\mathbf{z})|^{-1}, \qquad (7)$$

where $\pi(\cdot)$ is a known distribution (a *base* distribution, e.g., Normal distribution), $f : \mathcal{X} \to \mathcal{X}$ is a bijective map, and $\mathbf{J}_f(\mathbf{z})$ denotes the Jacobian matrix.

The main challenge of flow-based generative models lies in formulating invertible transformations for which the Jacobian determinant is computationally tractable. In the simplest case, we can use volume-preserving transformations that result in $\mathbf{J}_f(\mathbf{z}) = 1$, e.g., linear autoregressive flows (Kingma et al., 2016; Tomczak & Welling, 2017) or Householder flows (Tomczak & Welling, 2016). However, the volume-preserving flows cannot model arbitrary distributions, therefore, non-linear transformations are preferable. In (Rezende & Mohamed, 2015; Berg et al., 2018; Hoogeboom et al., 2020) a specific form of non-linear transformations was constructed so that to take advantage of the matrix determinant lemma and its generalization to efficiently calculate the Jacobian determinant. Recently, the transformation used in (Rezende & Mohamed, 2015; Berg et al., 2018) was further generalized to arbitrary contractive residual networks (Chen et al., 2019) and contractive densenets (Perugachi-Diaz et al., 2021) with the *Russian roulette estimator* of the Jacobian determinant. Coupling layers constitute a different group of non-linear transformation that are used in flow-based models like RealNVP (Dinh et al., 2016) and GLOW (Kingma & Dhariwal, 2018).

In the case of discrete variables, e.g., $\mathcal{X} = \mathbb{Z}^D$, the change of variables takes a simpler form due to the fact that there is no change of volume for probability mass functions:

$$p(\mathbf{x}) = \pi(\mathbf{z} = f^{-1}(\mathbf{x})). \qquad (8)$$

To date, coupling layers with the rounding operator (Eq. 6) are typically used. The resulting flow-based models are called Integer Discrete Flows (IDF) (Hoogeboom et al., 2019; Berg et al., 2020) with a mixture of discretized logistic distributions (Salimans et al., 2017) as the base distribution.

## 2. Our approach

### 2.1. General invertible transformations

Our main contribution of this paper is a proposition of a new class of invertible transformations that generalize many invertible transformations in reversible computing and reversible deep learning.

**Proposition 1.** *Let us take* $\mathbf{x}, \mathbf{y} \in \mathcal{X}$. *If binary transformations* $\circ$ *and* $\triangleright$ *have inverses* $\bullet$ *and* $\blacktriangleleft$, *respectively, and* $g_2, \ldots, g_D$ *and* $f_1, \ldots, f_D$ *are arbitrary functions, where* $g_i : \mathcal{X}_{1:i-1} \to \mathcal{X}_i$, $f_i : \mathcal{X}_{1:i-1} \times \mathcal{X}_{i+1:n} \to \mathcal{X}_i$, *then the following transformation from* $\mathbf{x}$ *to* $\mathbf{y}$:

$$
\begin{aligned}
y_1 &= x_1 \circ f_1(\emptyset, \mathbf{x}_{2:D}) \\
y_2 &= (g_2(y_1) \triangleright x_2) \circ f_2(y_1, \mathbf{x}_{3:D}) \\
&\cdots \\
y_d &= (g_d(\mathbf{y}_{1:d-1}) \triangleright x_d) \circ f_d(\mathbf{y}_{1:d-1}, \mathbf{x}_{d+1:D}) \\
&\cdots \\
y_D &= (g_D(\mathbf{y}_{1:D-1}) \triangleright x_D) \circ f_D(\mathbf{y}_{1:D-1}, \emptyset)
\end{aligned}
$$

*is invertible.*

*Proof.* In order to inverse $\mathbf{y}$ to $\mathbf{x}$ we start from the last element to obtain the following:

$$x_D = g_D(\mathbf{y}_{1:D-1}) \blacktriangleleft (y_D \bullet f_D(\mathbf{y}_{1:D-1}, \emptyset)).$$

Then, we can proceed with next expressions in the decreasing order (*i.e.*, from $D-1$ to 1) to eventually obtain:

$$
\begin{aligned}
x_{D-1} &= g_{D-1}(\mathbf{y}_{1:D-2}) \blacktriangleleft (y_{D-1} \bullet f_{D-1}(\mathbf{y}_{1:D-2}, x_D)) \\
&\cdots \\
x_d &= g_d(\mathbf{y}_{1:d-1}) \blacktriangleleft (y_d \bullet f_d(\mathbf{y}_{1:d-1}, \mathbf{x}_{d+1:D})) \\
&\cdots \\
x_2 &= g_2(y_1) \blacktriangleleft (y_2 \bullet f_2(y_1, \mathbf{x}_{3:D})) \\
x_1 &= y_1 \bullet f_1(\emptyset, \mathbf{x}_{2:D}).
\end{aligned}
$$

$\square$

Next, we show that many widely known invertible transformations could be derived from the proposed general invertible transformation. First, for the space of binary variables, we present that our proposition could be used to obtain three of the most important reversible logic gates.

**Corollary 2** (Feynman gate). *Let us consider* $\mathbf{x} \in \{0,1\}^2$, *and* $\circ \equiv \oplus$ *and* $\triangleright \equiv \oplus$ *with* $\bullet \equiv \oplus$ *and* $\blacktriangleleft \equiv \oplus$, *where* $\oplus$ *is the XOR operation. Then, taking* $g_2 \equiv 0$, $f_1(x_2) = 0$ *and* $f_2(y_1) = y_1$ *results in the Feynman gate:*

$$
\begin{aligned}
y_1 &= x_1 \\
y_2 &= x_2 \oplus x_1.
\end{aligned} \qquad (9)
$$

*Proof.* The Eq. 9 follows from the idempotency of XOR.

$\square$

**Corollary 3** (Toffoli gate). *Let us consider* $\mathbf{x} \in \{0,1\}^3$, *and* $\circ \equiv \oplus$ *and* $\triangleright \equiv \oplus$ *with* $\bullet \equiv \oplus$ *and* $\blacktriangleleft \equiv \oplus$, *where* $\oplus$ *is the XOR operation. Then, taking* $g_2(y_1) \equiv 0$, $g_3(\mathbf{y}_{1:2} \equiv 0,$

$f_1(\mathbf{x}_{2:3}) \equiv 0$, $f_2(y_1, x_3) \equiv 0$ and $f_3(\mathbf{y}_{1:2}) = y_1 \wedge y_2$ results in the Toffoli gate:

$$y_1 = x_1 \tag{10}$$
$$y_2 = x_2 \tag{11}$$
$$y_3 = x_3 \oplus (y_1 \wedge y_2). \tag{12}$$

*Proof.* The Eqs. 10 - 12 follow from the idempotency of the XOR operator. □

**Corollary 4** (Fredkin gate). *Let us consider* $\mathbf{x} \in \{0, 1\}^4$, *and* $\circ \equiv \oplus$ *and* $\triangleright \equiv \oplus$ *with* $\bullet \equiv \oplus$ *and* $\blacktriangleleft \equiv \oplus$, *where* $\oplus$ *is the XOR operation. Then, taking* $x_1 \equiv 0$, $g_2(y_1) \equiv 0$, $g_3(\mathbf{y}_{1:2}) \equiv 0$, $g_4(\mathbf{y}_{1:3}) \equiv 0$, $f_1(\mathbf{x}_{2:4}) = x_2 \wedge (x_3 \oplus x_4)$, $f_2(y_1, \mathbf{x}_{3:4}) \equiv 0$, $f_3(\mathbf{y}_{1:2}, x_4) = y_1$ *and* $f_4(\mathbf{y}_{1:3}) \equiv y_1$ *results in the Fredkin gate:*

$$y_1 = x_1 \oplus (x_2 \wedge (x_3 \oplus x_4)) \tag{13}$$
$$y_2 = x_2 \oplus 0 \tag{14}$$
$$y_3 = x_3 \oplus y_1 \tag{15}$$
$$y_4 = x_4 \oplus y_1 \tag{16}$$

**Remark 5** (On the Fredkin gate). *Comparing equations 13–16 with the definition of Fredkin gate we notice that in Corollary 4 we have to introduce an additional equation to be consistent with the Proposition 1. Moreover, we introduced a dummy variable* $x_1$ *that always equals* 0.

Moreover, we observe that our proposition generalizes invertible layers in neural networks.

**Corollary 6** (A coupling layer). *Let us consider* $\mathbf{x} = [\mathbf{x}_1, \mathbf{x}_2]^\top$, *where* $\mathcal{X}_i = \mathbb{R}^{D_i}$, *and* $\circ \equiv +$, $\triangleright \equiv \odot$ *with* $\bullet \equiv -$ *and* $\blacktriangleleft \equiv \oslash$, *where* $\odot$ *and* $\oslash$ *denote element-wise multiplication and division, respectively. Then, taking* $g_2(\mathbf{y}_1) = \exp(\mathrm{NN}_s(\mathbf{y}_1))$, $f_1(\mathbf{x}_2) = 0$ *and* $f_2(\mathbf{y}_1) = \mathrm{NN}_t(\mathbf{y}_1)$, *where* $\mathrm{NN}_s$, $\mathrm{NN}_t$ *are neural networks, results in the coupling layer (Dinh et al., 2016):*

$$\mathbf{y}_1 = \mathbf{x}_1 \tag{17}$$
$$\mathbf{y}_2 = \exp(\mathrm{NN}_s(\mathbf{y}_1)) \odot \mathbf{x}_2 + \mathrm{NN}_t(\mathbf{y}_1). \tag{18}$$

**Corollary 7** (A reversible residual layer). *Let us consider* $\mathbf{x} = [\mathbf{x}_1, \mathbf{x}_2]^\top$, *where* $\mathcal{X}_i = \mathbb{R}^{D_i}$, $\circ \equiv +$, $\triangleright \equiv \odot$ *with* $\bullet \equiv -$ *and* $\blacktriangleleft \equiv \oslash$, *where* $\odot$ *and* $\oslash$ *denote element-wise multiplication and division, respectively. Then, taking* $g_2(\mathbf{y}_1) \equiv 1$, $f_1(\mathbf{x}_2) = \mathrm{NN}_1(\mathbf{x}_2)$ *and* $f_2(\mathbf{y}_1) = \mathrm{NN}_2(\mathbf{y}_1)$, *where* $\mathrm{NN}$ *is a neural network, results in the reversible residual layer proposed in (Gomez et al., 2017):*

$$\mathbf{y}_1 = \mathbf{x}_1 + \mathrm{NN}_1(\mathbf{x}_2) \tag{19}$$
$$\mathbf{y}_2 = \mathbf{x}_2 + \mathrm{NN}_2(\mathbf{y}_1). \tag{20}$$

**Remark 8** (On the reversible residual layer). *According to Proposition 1, we can further generalize the reversible residual layer proposed in (Gomez et al., 2017) by taking* $g_2(\mathbf{y}_1) = \exp(\mathrm{NN}_3(\mathbf{y}_1))$ *that would result in the following invertible layer:*

$$\mathbf{y}_1 = \mathbf{x}_1 + \mathrm{NN}_1(\mathbf{x}_2) \tag{21}$$
$$\mathbf{y}_2 = \exp(\mathrm{NN}_3(\mathbf{y}_1)) \odot \mathbf{x}_2 + \mathrm{NN}_2(\mathbf{y}_1). \tag{22}$$

*Interestingly, we can calculate the Jacobian of such transformation that takes the following form:*

$$\mathbf{J}(\mathbf{z}) = \begin{bmatrix} \frac{\partial \mathbf{y}_1}{\partial \mathbf{x}_1} & \frac{\partial \mathbf{y}_1}{\partial \mathbf{x}_2} \\ \frac{\partial \mathbf{y}_2}{\partial \mathbf{x}_1} & \frac{\partial \mathbf{y}_2}{\partial \mathbf{x}_2} \end{bmatrix} = \begin{bmatrix} \mathbf{A} & \mathbf{B} \\ \mathbf{C} & \mathbf{D} \end{bmatrix} \tag{23}$$

*where* $\mathbf{A} = \mathbf{I}$, *i.e., the identity matrix. Then,* $\mathbf{A}\mathbf{C} = \mathbf{C}\mathbf{A}$ *and according to Theorem 3 on determinants of block matrices in (Silvester, 2000), the logarithm of the Jacobian-determinant equals:*

$$\log |\det \mathbf{J}(\mathbf{z})| = \sum_i \mathrm{NN}_{3,i}(\mathbf{y}_1). \tag{24}$$

**Corollary 9** (A reversible differential mutation). *Let us consider* $\mathbf{x}_1, \mathbf{x}_2, \mathbf{x}_3 \in \mathbb{R}^D$, $\gamma \in \mathbb{R}_+$, *and* $\circ \equiv +$, $\triangleright \equiv \odot$ *with* $\bullet \equiv -$. *Then, taking* $g_2(\mathbf{y}_1) \equiv 1$, $g_3(\mathbf{y}_{1:2}) \equiv 1$, $f_1(\mathbf{x}_{2:3}) = \gamma(\mathbf{x}_2 - \mathbf{x}_3)$, $f_2(\mathbf{y}_1, \mathbf{x}_3) = \gamma(\mathbf{x}_3 - \mathbf{y}_1)$, *and* $f_3(\mathbf{y}_{1:2}) = \gamma(\mathbf{y}_1 - \mathbf{y}_2)$, *results in the reversible differential mutation proposed in (Tomczak et al., 2020):*

$$\mathbf{y}_1 = \mathbf{x}_1 + \gamma(\mathbf{x}_2 - \mathbf{x}_3) \tag{25}$$
$$\mathbf{y}_2 = \mathbf{x}_2 + \gamma(\mathbf{x}_3 - \mathbf{y}_1) \tag{26}$$
$$\mathbf{y}_3 = \mathbf{x}_3 + \gamma(\mathbf{y}_1 - \mathbf{y}_2). \tag{27}$$

## 2.2. General Invertible Transformations for Integer Discrete Flows

We propose to utilize our general invertible transformations in IDF. For this purpose, we formulate two new coupling layers that fulfill Proposition 1, namely:
– We divide the input into four parts, $\mathbf{x} = [\mathbf{x}_1, \mathbf{x}_2, \mathbf{x}_3, \mathbf{x}_4]$:

$$\begin{aligned} \mathbf{y}_1 &= \mathbf{x}_1 + \lfloor \mathrm{NN}_{t,1}(\mathbf{x}_{2:4}) \rceil \\ \mathbf{y}_2 &= \mathbf{x}_2 + \lfloor \mathrm{NN}_{t,2}(\mathbf{y}_1, \mathbf{x}_{3:4}) \rceil \\ \mathbf{y}_3 &= \mathbf{x}_3 + \lfloor \mathrm{NN}_{t,3}(\mathbf{y}_{1:2}, \mathbf{x}_4) \rceil \\ \mathbf{y}_4 &= \mathbf{x}_4 + \lfloor \mathrm{NN}_{t,4}(\mathbf{y}_{1:3}) \rceil \end{aligned} \tag{28}$$

– We divide the input into eight parts, $\mathbf{x} = [\mathbf{x}_1, \ldots, \mathbf{x}_8]$. Then, we formulate the coupling layer analogically to (28).

Further, we use the discretized two-parameter logistic distribution. It could be expressed as a difference of the logistic cumulative distribution functions (Salimans et al., 2017) or analytically (Chakraborty & Chakravarty, 2016).

## 3. Experiments

**Data** In the experiment, we use a toy dataset of handwritten digits available in Scikit-learn[1] that consists of $1,797$ images. Each image consists of 64 pixels (8px×8px).

**Models** In the experiment, we compare the following models: RealNVP with uniform dequantization and the standard Gaussian base distribution (REALNVP), RealNVP with the coupling layer in Remark 8 and the standard Gaussian base distribution (REALNVP2), IDF with the coupling layer in (6) (REALNVP), IDF with the coupling layer in (28) (IDF4), IDF with the coupling layer for eight parts (IDF8). In all models, we used the order permutation (i.e., a matrix with ones on the anti-diagonal and zeros elsewhere) after each coupling layer.

In order to keep a similar number of weights, we used 16 flows for IDF, 8 flows for REALNVP, 4 flows for REAL-NVP2, 4 flows for IDF4, and 2 flows for IDF8. All models have roughly $1.32$M weights. For IDF, IDF4, and IDF8 we utilized the following neural networks for transitions $(NN_t)$:

$$\text{Linear}(D_{in}, 256) \rightarrow \text{LeakyReLU} \rightarrow \text{Linear}(256, 256) \rightarrow$$

$$\text{LeakyReLU} \rightarrow \text{Linear}(256, D_{out})$$

and for REALNVP, we additionally used the following neural networks for scaling $(NN_s)$:

$$\text{Linear}(D_{in}, 256) \rightarrow \text{LeakyReLU} \rightarrow \text{Linear}(256, 256) \rightarrow$$

$$\text{LeakyReLU} \rightarrow \text{Linear}(256, D_{out}) \rightarrow \text{Tanh}$$

**Training & Evaluation** We compare the models using the negative-log likelihood (nll). We train each model using $1,000$ images, and the mini-batch equals 64. Moreover, we take 350 images for validation and 447 for testing. Each model is trained 5 times. During training, we use the early stopping with the patience equal 20 epochs, and the best performing model on the validation set is later evaluated on the test set. The Adam optimizer with the learning rate equal 0.001 was used.

**Code** The experiments could be reproduced by running the code available at: https://github.com/jmtomczak/git_flow

**Results** In Figure 1, we present aggregated results for the five models. Moreover, in Figure 2, examples of unconditional samples are presented, together with a real sample.

First, we notice that IDF performs better than REALNVP and REALNVP2. In this paper, we use fully-connected neural networks and small toy data. Nevertheless, it is interesting to see that IDF could perform better than a widely used

---

[1] https://scikit-learn.org/stable/datasets/index.html#digits-dataset

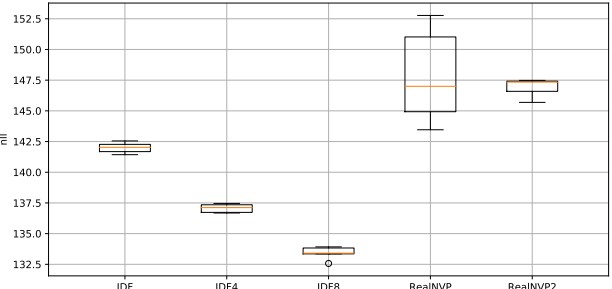

*Figure 1.* The aggregated results for the four models on the test set.

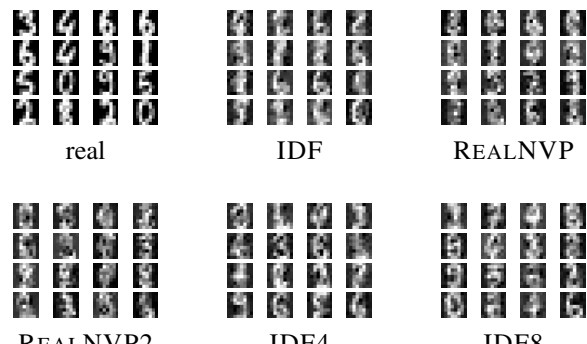

*Figure 2.* Unconditional samples from the five models.

continuous flow-based model with dequantization. Moreover, it seems that the new coupling layer presented in Remark 8 is more stable in terms of final results, however, the difference in performance is statistically insignificant. Second, we observe that our proposition of invertible transformations results in improved nll. The proposed coupling layer with 8 partitions performed the best in terms of nll, and the coupling layer with 4 partitions also outperformed IDF, REALNVP and REALNVP2. We want to highlight that all models have almost identical numbers of weights, thus, these results are not caused by taking larger neural networks. The samples presented in Figure 2 are rather hard to analyze due to their small size (8px×8px). Nevertheless, we notice that all IDFs generated crisper images than REALNVP and REALNVP2. Moreover, it seems that IDF4 and IDF8 seem to produce digits of higher visual quality.

## 4. Conclusion

In this paper, we proposed a new class of invertible transformations. We showed that many well-known invertible transformations could be derived from our proposition. Moreover, we proposed two coupling layers and presented how they could be utilized in flow-based models for integer values (Integer Discrete Flows). Our preliminary experiments on the digits data indicate that the new coupling layers result in better negative log-likelihood values than for IDF and REALNVP. These results are promising and will be pursued in the future on more challenging datasets.

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
