# OpenReview forum: "General Invertible Transformations for Flow-based Generative Modeling"
_ICML.cc/2021/Workshop/INNF — INNF+ 2021 poster_

### Official Review · Reviewer_nfm8 · 2021-06-08

**Rating:** Borderline Accept
**Confidence:** 5

**Summary:**

This paper proposes a new class of invertible transformations inspired by reversible logic and computing. The proposed class is shown to be able to capture many canonical flows such as Coupling Flows and Integer Discrete Flows.


**Justification For Rating:**

Overall, I found this paper to be very well written and I found the historical perspective with reversible logic gates to be an interesting connection. I think providing connections to different areas of research---especially ones that lead to new insights is always illuminating and welcome. Having said that I do have some questions regarding the two new proposed flows. It seems like you must have an input dimensionality >=4 and >=8 respectively? Unless I'm mistaken these proposed flows would not be able to model toy data such as a mixture of 2D gaussians. If this is indeed a limitation then it should be stated in the paper. From the experiments standpoint why not just use Binarized-MNIST instead of a downsampled version of MNIST? Also, do the reported NLL scores use the IWAE bound? If not I would encourage the authors to do it as it will only improve results. Another useful baseline that could be worth comparing with is the Autoregressive flow or Neural Autoregressive flow as it's similar in spirit to proposition one in that we model each dimension sequentially.

Minor Comments:
- In Eqn 5, you also have to remove the exp transform. In terms of notation, what does NN_{t,1} mean?
- Missing citation of the straight-through estimator
- There are actually more than 3 major groups of generative models. For example, score-based generative models.
- Missing citation for Discrete: "Discrete flows: Invertible generative models of discrete data" by Tran et. al and "A RAD approach to deep mixture models" by Dinh et. al
- How do you propose to divide the input if the dimensionality is less than 4 and 8 respectively?

---

### Official Review · Reviewer_yJzF · 2021-06-11

**Rating:** Borderline Reject
**Confidence:** 4

**Summary:**

The paper attempts to introduce a "generalized" definition of the coupling layer. The paper emphasizes that we can design an invertible function by using a binary operator and fixing the other argument; for example, the addition of a real value to input. Nothing this, the author proposes to define generalized coupling layers: here, an invertible function on a subset of input will be defined by a binary operator conditioning on the remaining input.

Using this framework, the author interprets reversible logic gates (aka invertible logic gates in the paper) as a special case, where "AND" and "XOR" binary operators are used.

In the experiments, the author introduces to increase partial dependencies of integer discrete flow (IDF), similarly to invertible autoregressive flow (IAF) against a coupling layer.

**Justification For Rating:**

Overall I like the general direction of the paper. The following aspects of the paper can be further improved.

1. It is unclear how we can "generalize" coupling layers, whilst it supposes to be the main contribution. Specifically, Proposition 1 is incomplete. It is unclear under which conditions of binary operators and the given elements, we can design invertible functions. For example, element-wise norm operations are also binary operators; in general, however, it is not invertible on one argument for fixing the other argument. Another counterexample includes the multiplication of 0 to an input; note that the exponential function is used in the element-wise multiplications in the coupling layers. Remind that the paper introduces some new cases, such as reversible logic gates. However, since the proposition doesn't specify anything about conditions, it is also unclear why the gates are spacial cases. \
In this perspective, I strongly believe that completing Proposition 1 by clarifying the conditions will be a great contribution to the flow-based model community. On the other hand, if one found that the reversible logic gates are interesting examples, it would be great to extend this direction as well. Can we define parametric reversible logic gates, similar to other flows? Will these parametric logic gates be trained efficiently? Will the composition of these reversible gates be universal?

2. I consider that the experiments are not relevant to the main contribution of the paper. The experiments demonstrate that increasing partial dependencies improves the expressivity of a single coupling layer by sacrificing computation time. I don't think it is a necessary experiment to support the main contribution if the contribution is a generalization. It would nicer if the learnable logic gates can be used in discretized modeling instead of IDF (if possible).

---

### Official Review · Reviewer_m9nR · 2021-06-11

**Rating:** Borderline Reject
**Confidence:** 4

**Summary:**

This paper proposes new coupling layers for use in Integer Discrete Flows, motivated by results from reversible computing. The idea is validated on a tiny non-standard MNIST-like dataset.

**Justification For Rating:**

The contribution is novel and potentially useful. It's a good fit for this workshop. The paper is wel written. However, the empirical results are extremely preliminary: the dataset that the authors experiment on is non-standard and very small in size: 1797 8x8 images. It is not clear how the method compares against well-tuned baselines. The authors do compare against their own implementation of some of these baselines, but it is impossible for me to assess how well these have been implemented. Although the bar on experiments should be low for workshop submissions, I feel like the authors should at least run their method on a more standard benchmark like MNIST. If resources are a constraint this can still be done with e.g. a free google GPU colab.

---

### Decision · Program_Chairs · 2021-06-14

**Decision:**

Accept (poster)

**Comment:**

The paper is a good fit for the workshop, and explores a set of new ideas. The reviewers pointed out that the experimentally evaluation is lacking, and that there might be gaps in the theory or at least in the presentation of it. Despite these shortcomings, we decided to accept the paper to encourage discussion of novel connections and ideas. However, we urge the authors to take into account the reviewers' criticism, and to consider ways the paper can be improved.